# Long-Term Mapping of a Greenhouse in a Typical Protected Agricultural Region Using Landsat Imagery and the Google Earth Engine

**Cong Ou [1,2]**, **Jianyu Yang [1,2,***, **Zhenrong Du [1,2]**, **Yiming Liu [1,2]**, **Quanlong Feng [1,2]** and **Dehai Zhu [1,2]**

[1] Department of Geographic Information Engineering, College of Land Science and Technology, China Agricultural University, No.17 Tsing Hua East Road, Haidian District, Beijing 100083, China; oucong@cau.edu.cn (C.O.); duzhenrong@cau.edu.cn (Z.D.); liuym0086@cau.edu.cn (Y.L.); fengql@cau.edu.cn (Q.F.); zhudehai@cau.edu.cn (D.Z.)

[2] Key Laboratory for Agricultural Land Quality Monitoring and Control, Ministry of Natural Resources, Beijing 100083, China

* Correspondence: ycjyyang@cau.edu.cn; Tel.: +86-136-930-30210

**Abstract:** The greenhouse is the fastest growing food production approach and has become the symbol of protected agriculture with the development of agricultural modernization. Previous studies have verified the effectiveness of remote sensing techniques for mono-temporal greenhouse mapping. In practice, long-term monitoring of greenhouse from remote sensing data is vital for the sustainable management of protected agriculture and existing studies have been limited in understanding its spatiotemporal dynamics. This study aimed to generate multi-temporal greenhouse maps in a typical protected agricultural region (Shouguang region, north China) from 1990 to 2018 using Landsat imagery and the Google Earth Engine and quantify its spatiotemporal dynamics that occur as a consequence of the development of protected agriculture in the study area. The multi-temporal greenhouse maps were produced using random forest supervised classification at seven-time intervals, and the overall accuracy of the results greater than 90%. The total area of greenhouses in the study area expanded by 1061.94 km$^2$ from 1990 to 2018, with the largest growth occurring in 1995–2010. And a large number of increased greenhouses occurred in 10–35 km northwest and 0–5 km primary roads buffer zones. Differential change trajectories between the total area and number of patches of greenhouses were revealed using global change metrics. Results of five landscape metrics showed that various landscape patterns occurred in both spatial and temporal aspects. According to the value of landscape expansion index in each period, the growth mode of greenhouses was from outlying to edge-expansion and then gradually changed to infilling. Spatial heterogeneity, which measured by Shannon's entropy, of the increased greenhouses was different between the global and local levels. These results demonstrated the advantage of utilizing Landsat imagery and Google Earth Engine for monitoring the development of greenhouses in a long-term period and provided a more intuitive perspective to understand the process of this special agricultural production approach than relevant social science studies.

**Keywords:** greenhouse; remote sensing; multi-temporal; spatiotemporal dynamics; Landsat imagery; Google Earth Engine

## 1. Introduction

One of the biggest challenges facing the future world is the need to increase the production of food to feed a growing population, a large part of which needs to rely on protected agriculture to

increase food supply [1]. As the most important symbol of protected agriculture, the greenhouse is a technology-based approach toward food production that can be characterized by strong risk resistance, large input of material and energy, highly concentrated knowledge and technology, distinct region discrepancy and multiple effects in the economy, society and ecology [2,3]. With rapid urbanization and the population explosion since the mid-1980s, the greenhouse has rapidly become a new industry with the development of agricultural modernization in China. According to the Second and Third National Agricultural Census in China, the entire country of China contained 7770 km$^2$ greenhouses in 2006 and 13,150 km$^2$ in 2016. The significant increment of greenhouses indicates that it not only changed the form of seasonal food production but also reshaped the landscape of farmland in suburban areas. Simultaneously, the expansion of greenhouses also results in some environmental problems, such as soil degradation after long-term fertilization [4] and agricultural wastes (vegetable, plastic, chemical, etc.) [5]. To alleviate these contradictions and maintain the balance between food supply and environmental security, the region planner should make exact judgments on the extent of greenhouses expansion. In this context, a better understanding of its spatiotemporal dynamics overtime is required in a country or in a given region. Furthermore, as a special food production method on cultivated land, multi-temporal greenhouse mapping can also provide basic geomatics information for other agricultural monitoring [6–10].

Remote sensing techniques have been widely applied for detecting the spatial distribution of greenhouses. Previous studies have focused on the topic of mono-temporal precise greenhouse mapping based on multiple types of satellite imagery, from mid-resolution images like Landsat TM/ETM+ [11], Sentinel-2 MSI [12] or GF-2 [13] to high-resolution images such as QuickBird [14] or WorldView-2 [15]. And in these aforementioned works, both pixel-based and object-based approaches have been performed. Regarding greenhouse mapping from the spectral characteristics, various indices such as the vegetable land extraction index (Vi) [16], moment distance index (MDI) [17], plastic-mulched landcover index (PMLI) [18], plastic greenhouses index (PGI) [19] and greenhouses detection index (GDI) [20] have been proposed. However, few researchers have addressed the topic of multi-temporal greenhouse mapping and the analysis of their dynamics due to data computation and storage limits. Therefore, it is difficult to attain a clear picture of greenhouses over a long historical period.

Long-term satellite images, such as the Landsat series, are becoming increasingly available and essential to help us understand the geospatial process. There are extensive applications of Landsat images for multi-temporal land use/cover mapping. Representative works include land cover and land-use changes [21], habitat and agricultural land cover mapping [22], urban expansion mapping [23], crop mapping [24] and forest disturbance and recovery detection [25], etc. In particular, since the USGS made their entire Landsat archive freely available to the public in 2008 [26], it provides an advantage opportunity to monitor successive greenhouse changes at considerable spatial and temporal resolutions. However, because the production of long-term greenhouse maps for large areas (>10,000 km$^2$) from Landsat images needs to store, manage and process a large amount of data, it still requires considerable high-performance computing resources to facilitate these data for arctic applications. Fortunately, a cloud-based platform named Google Earth Engine (GEE) was launched by Google in December 2010. This geospatial analysis platform makes it easy to process very large geospatial datasets with a high-performance, intrinsically parallel computation service [27]. Meantime, it contains more than forty years of Earth-observing remote sensing imagery and all of this data is preprocessed so that the researchers could analyze real-time changes to the Earth's surface without many barriers associated with data management.

In summary, multi-temporal greenhouse mapping and the analysis of the associated dynamics is crucial to understand and assess the sustainable development of protected agriculture and provides a complete representation and patterns of protected agriculture that exist in space and time. In monitoring greenhouses from satellite imagery, current studies tend to introduce many varied methods to improve classification accuracy and ignore their dynamics in a long-term period. Therefore, the

overarching purpose of this study is to develop and test advanced image classification techniques with Landsat time-series in GEE for multi-temporal greenhouse mapping in a typical protected agriculture region and quantify their spatiotemporal characteristics. The specific objectives of this study were to: (1) generate accurate maps of greenhouses in Shouguang region using Landsat imagery and the Google Earth Engine; (2) quantify spatiotemporal dynamics characteristics of greenhouses during 1990–2018 at the global and local levels by using global change metrics, landscape pattern metrics, landscape expansion index and spatial entropy measure. In addition, compared with relevant social science studies like the agricultural extension [28] or agricultural economic development [29] based on statistical data or questionnaire surveys, our research based on long-term remote sensing data contains geographical information and may be able to provide a more intuitive perspective for similar studies.

## 2. Materials and Methods

### 2.1. Study Area

The study area is located in the northern part of Shandong Province, China (36.058°~37.362°N and 118.113°~119.575°E). It is a suburban area neighboring cities of Zibo and Weifang, which is approximate 12,646 km$^2$ and contains a population of approximately 5.84 million. It is an important agricultural zone in Shandong, with administrative areas including Guangrao, Linzi, Qingzhou, Weicheng, Changle, Kuiwen, Fangzi, Linqu, Anqiu, Shouguang, and Hanting (as shown in Figure 1). This area is dominated by three landforms: hills, plains and coastal areas and the hypsography of this area is southern high and northern Low-Lyin. The study area belongs to the north temperate monsoon area under the joint influence of Eurasia and the Pacific Ocean, characterized by a dry spring, hot and rainy summer, cool autumn, and dry and cold winter. The annual average temperature of this area is 12.7 °C and the average total sunshine hours are 2548.8 h, which favors agricultural production.

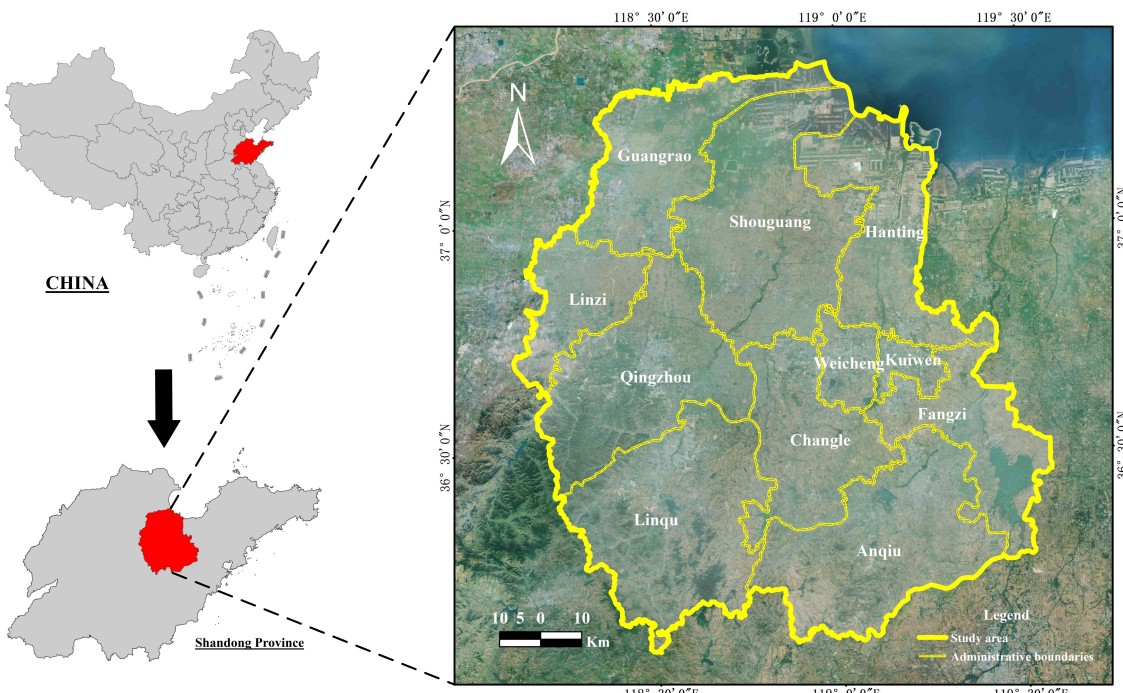

**Figure 1.** Location of the study area.

Due to its climatic characteristics and geographical location, the study area we selected has the largest vegetable production and wholesale market in China, and the so-called 'China's Vegetable Township'. Among the 11 counties mentioned above, Shouguang, in particular, is the cradle of 'winter-warm' greenhouses and the largest vegetable production base in China. Considering the latest

distribution of greenhouses in this area, we selected the above counties as a typical region in order to trace the full change trajectory of greenhouses. In addition, according to some historical records, the large-scale specialized greenhouses planting in this area began in the early 1990s, thus in this study, we developed the multi-temporal greenhouses maps based on the 1990–2018 period with a five-year interval.

## 2.2. Data Acquisition and Preprocessing

Data catalog in GEE contains archived Landsat data in its Raw Images (representing scaled, calibrated at-sensor radiance), TOA (representing calibrated top-of-atmosphere (TOA) reflectance) and SR (representing atmospherically corrected surface reflectance) forms, and includes Landsat 5 from 1984 to 2012, Landsat 7 from 1999 to present, and Landsat 8 from 2013 to present. Landsat imagery scenes were used in this study for the years 1990, 1995, 2000, 2005, 2010, 2015 and 2018, and the extent of the study area was covered by 6 scenes of Landsat images (paths 120–122, rows 34–35) for each year. Due to the SLC-off issue (when the Scan Line Corrector failed and these products have data gaps) of Landsat 7 after 31 May 2003 [30], we chose Landsat 5 Surface Reflectance (SR) for the period 1990–2010 and Landsat 8 Surface Reflectance (SR) for period 2015–2018. The data processing has four parts: first, considering the phenological characteristics of plant growth in the greenhouses, four layers that covers the study area were chosen for each year: spring season layer (1 March to 31 May), summer season layer (1 June to 31 August), autumn season layer (1 September to 30 November) and winter season layer (1 December to 28/29 February) to enhance the reproducibility of the classification. Then, an algorithm called CFmask [31] based on the "pixel_qa" (pixel quality attributes) band of Landsat SR data was applied to mask clouds and create cloud- and cloud-shadow- free Landsat images, and these images were clipped by the boundary of the study area. After that, a time series of Landsat image composites in each season was reduced by calculating the mean of all values at each pixel across the stack of all matching bands. Finally, three visible (RGB) and near-infrared bands were selected for basic classification features. In order to reduce the noise resulting from different sensors and images, the Normalized Difference Vegetation Index (NDVI) [32], Normalized Difference Built Index (NDBI) [33], Modified Normalized Difference Water Index (MNDWI) [34] were also performed for later classification. Besides that, since the greenhouses are not likely to appear in the mountainous areas with steep slopes, but mainly distributed in the plain farming areas. Therefore, Elevation and slope, which are derived from Shuttle Radar Topography Mission (SRTM) data [35], were used to distinguish some misclassified objects like white rural buildings in mountainous areas.

## 2.3. Reference Dataset for Training Samples

The maps of greenhouses for 1990, 1995, 2000, 2005, 2010, 2015 and 2018 were derived separately using a supervised machine learning approach at 30-m spatial resolution. The reference dataset is an important consideration to the accuracy of classification results [36], which was divided into two parts: the training and validation datasets. The training dataset was used to train the supervised classifier, while the validation dataset was used in accuracy assessment of the produced greenhouse classification results. Field surveys were conducted from 2017 to 2019 along the study area to collect ground truth data. A total of 1162 sampling polygons were drawn according to the location points and higher-resolution imagery, including 658 polygons of greenhouses and 504 polygons of others (refers to non-greenhouse objects). Then we used the asset manager to upload sampling polygons in the Shapefile format via Google Fusion Tables and generated 10,000 points that were uniformly random within the given polygons by the "randomPoints" function in GEE. Meanwhile, 70% of these points were randomly selected as the training points and the remaining 30% were randomly selected as the validation points to evaluate the accuracy of classification results in 2018. Due to the lack of field survey data in 1990, 1995, 2000, 2005, 2010, and 2015, we collected 2490 sampling polygons via visual inspection of high-resolution imagery such as QuickBird and IKONOS available in Google Earth or Landsat images aided by previous knowledge such as greenhouse address records from local historical

agricultural statistics. Finally, the reference dataset obtained in 1990, 1995, 2000, 2005, 2010, and 2015 consisted of 123, 353, 444, 545, 469 and 556 sampling polygons of greenhouses, respectively and were also divided into the training and validation points like what we dealt with the sampling polygons in 2018 (as shown in Figure 2).

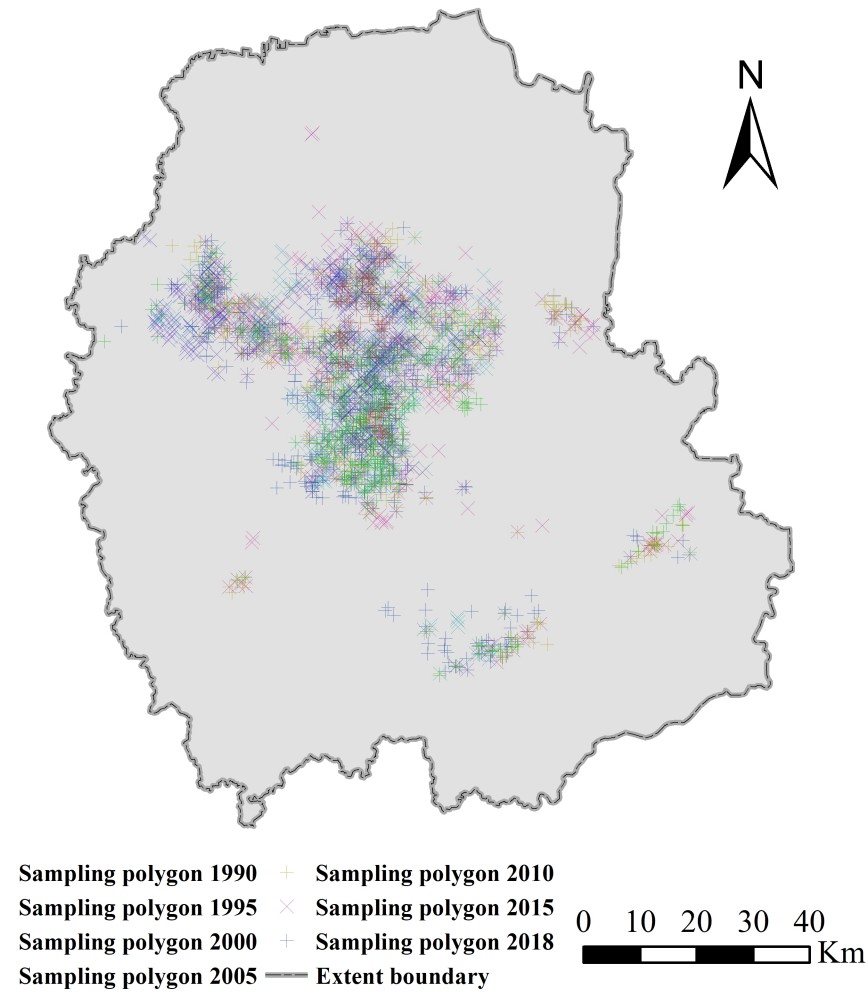

| | | | |
|---|---|---|---|
| + | **Sampling polygon 1990** | + | **Sampling polygon 2010** |
| × | **Sampling polygon 1995** | × | **Sampling polygon 2015** |
| + | **Sampling polygon 2000** | + | **Sampling polygon 2018** |
| × | **Sampling polygon 2005** | — | **Extent boundary** |

0  10  20  30  40 Km

**Figure 2.** Sampling polygons (coverted into point for a better visualization) in seven intervals.

*2.4. Image Classification and Accuracy Assessments*

Supervised machine learning uses data samples or past experiences to train the computer to optimize classification results [37], and a series of such algorithms have been used to remotely sensed data classification [38]. Six supervised machine learning algorithms are provided by GEE as follows: Classification and Regression Trees (CART) [39], Decision Tree (DT) [40], GMO Maximum Entropy [41], Naive Bayes [42], Random Forest (RF) [43], Support Vector Machine (SVM) [44]. Previous studies have verified the effectiveness of the RF algorithm in the classification accuracy due to the superior performance of its multi-dimensional features [45–47]. Therefore, we chose the RF algorithm as a classifier with ensembles of 500 trees to obtain the greenhouses classification maps for each chosen year. In addition, the accuracy assessment is a need for determining how well the classifier trained [48]. Four indices were used as the criteria to evaluate the produced greenhouses classification maps for each year in this study, including overall accuracy, kappa coefficient, producer's accuracy and consumer's accuracy. The general procedures of mapping greenhouses in the study area from 1990 to 2018 using the workflow [49] illustrated in Figure 3.

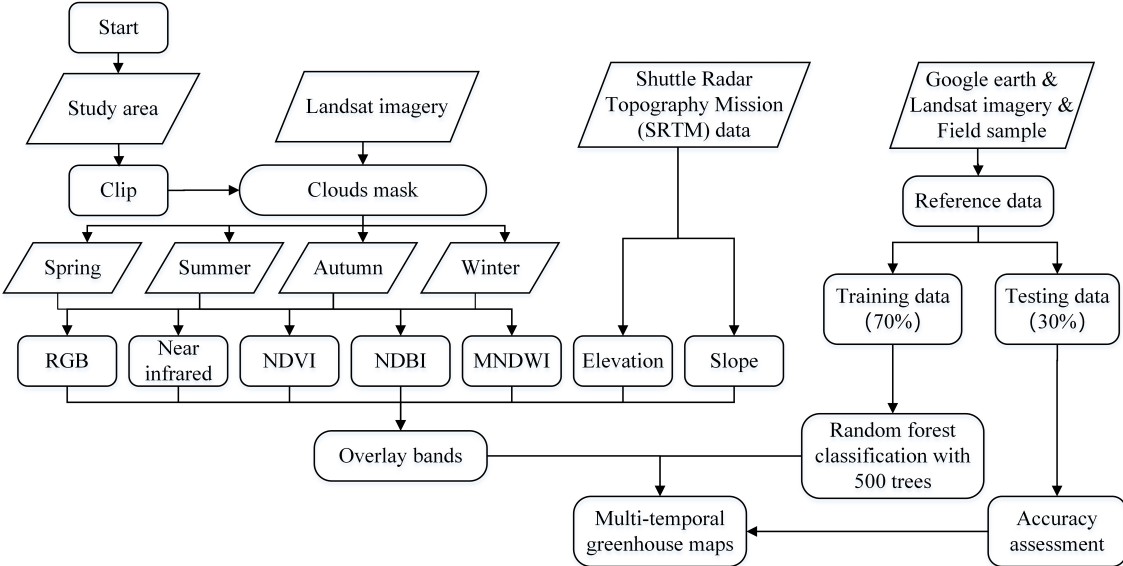

**Figure 3.** Flow chart of developing multi-temporal greenhouses maps.

## 2.5. Area Changes Analysis

As discussed previously, multi-temporal greenhouse maps have great potential in revealing long term dynamics of greenhouses extents. We analyzed the spatiotemporal dynamics of the greenhouses at the global and local levels, including Distance-based and Direction-based buffer zones. The spread of greenhouses is a site selection result of agricultural producers according to market demand, policy planning, transportation conditions and natural endowment. In order to reveal the influence of local differences in the process of greenhouse expansion, four different Distance-based buffer zones of rural settlements (DIST_RS), town centers (DIST_TC), primary roads (DIST_PR) and main rivers (DIST_MR) were created for further analysis (Table 1 and Figure 4). The shapefiles of rural settlements and town centers in the study area were provided by Data Center for Resources and Environmental Sciences, Chinese Academy of Sciences (http://www.resdc.cn), and the shapefiles of primary roads and main rivers in the study area were derived from OpenStreetMap (http://www.openstreetmap.org). Further, the pattern of greenhouse expansion has not been uniform in all directions. To understand the pattern of greenhouse growth in different directions, the study area has been divided into four Direction-based buffer zones based on Northwest (DIRE_NW), Northeast (DIRE_NE), Southwest (DIRE_SW) and Southeast (DIRE_SE), respectively (Figure 4), and each zone was divided into concentric circle of incrementing radius of 5 km from the central pixel of the study area.

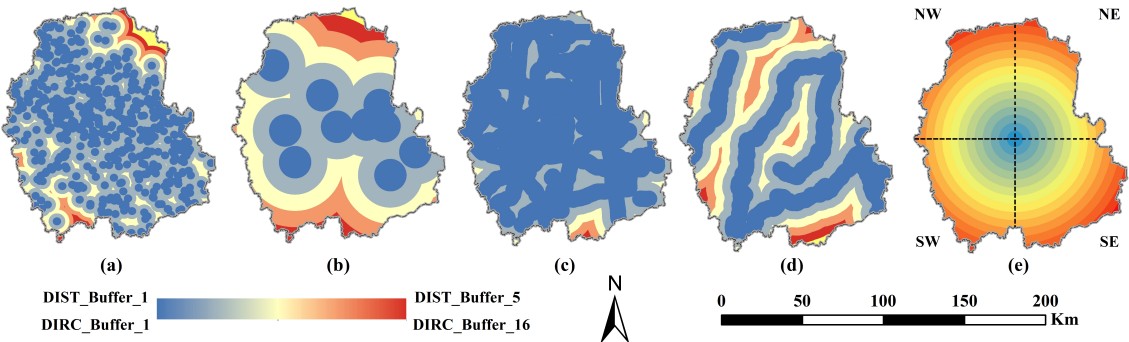

**Figure 4.** Distance-based and Direction-based buffer zones: (**a**) rural settlements, (**b**) town centers, (**c**) primary roads, (**d**) main rivers, and (**e**) 4 buffer zones based on directions (Northwest (NW), Northeast (NE), Southwest (SW), and Southeast (SE)).

**Table 1.** Buffer zones of rural settlements, town centers, primary roads and main rivers.

|  | Buffer_1 (km) | Buffer_2 (km) | Buffer_3 (km) | Buffer_4 (km) | Buffer_5 (km) |
|---|---|---|---|---|---|
| Rural settlements (DIST_RS) | 0–2.5 | 2.5–5 | 5–7.5 | 7.5–10 | 10–12.5 |
| Town centers (DIST_TC) | 0–10 | 10–20 | 20–30 | 30–40 | 40–50 |
| Primary roads (DIST_PR) | 0–5 | 5–10 | 10–15 | 15–20 | 20–25 |
| Main rivers (DIST_MR) | 0–5 | 5–10 | 10–15 | 15–20 | 20–25 |

The total area and the annual change rate ($km^2/year$) of the greenhouses were calculated for each interval at the global level, and the net increased greenhouse areas from 1990 to 2018 was monitored through the computation of overlap analysis at the local level. The annual change rate (ACR) of greenhouses is defined as follows:

$$ACR = (G_{end} - G_{start})/n \tag{1}$$

where $G_{end}$ and $G_{start}$ represents the total area of greenhouses at the end and start of the period, respectively, and $n$ is the duration between the start and end time.

For multi-temporal geographic data, it is essential to describe global changes explicitly through time. In this study, three spatially global change metrics were calculated using STAMPR [50,51] for a quantitative approach to the description of spatial changes in temporal neighbors: the number ratio (NumRatio), the area ratio (AreaRatio), and the average area ratio (AvgAreaRatio). These that can be expressed by:

$$NumRatio = \frac{N(t_1)}{N(t_2)} \tag{2}$$

$$AreaRatio = \frac{A(t_1)}{A(t_2)} \tag{3}$$

$$AvgAreaRatio = \frac{AreaRatio}{NumRatio} \tag{4}$$

where $N(t_1)$ and $N(t_2)$ represents the number of greenhouse patches in time interval one ($t_1$) and time interval two ($t_2$), respectively, and $A(t_1)$ and $A(t_2)$ represents the area of greenhouse patches in $t_1$ and $t_2$, respectively. The NumRatio measures the change in the number of greenhouse patches over two time periods, the AreaRatio measures the change in the area of greenhouse patches over two time periods, and the AvgAreaRatio normalizes the AreaRatio by the NumRatio.

*2.6. Landscape Pattern Change Analysis*

Following the produced multi-temporal greenhouse maps, we calculated landscape metrics for each of the maps to explore landscape pattern change of the greenhouses. Five landscape metrics were computed for each map at the landscape level with the help of FRAGSTATS [52,53] in the following: the number of greenhouse patches (NP) [54], edge density (ED) [54], landscape shape index (LSI) [55], area-weighted mean patch fractal dimension (AWMPFD) [56], and aggregation index (AI) [57]. Among them, NP and ED can be used to evaluate the fragmentation of the greenhouses, AWMPFD and LSI can be used to measure the complexity of the greenhouses, and AI can be used to assess the compactness of the greenhouses. Although some of them are highly correlated and only a few have been shown to be uncorrelated, simultaneous consideration of a collection of metrics is still required regarding the complex patterns of greenhouse expansion. In addition to calculating the landscape change pattern of the entire study area at each interval, we also analyzed the trends of these landscape indices in each Distance-based and Direction-based buffer zone based on the increased greenhouse map from 1990 to 2018.

## 2.7. Spatial Modes of Landscape Expansion

Given a greenhouse change rate, its changes may assume different expansion modes such as infilling, edge-expansion and outlying. To quantify these growth modes and capture the information of the formation process of landscape patterns, we used landscape expansion index (LEI) and its variants in this study [58]. LEI can be formulated as follows:

$$LEI = 100 \times \frac{A_0}{A_0 + A_v} \tag{5}$$

where $A_0$ is the intersection between a predefined buffer around a new greenhouse patch and previously existing greenhouse patch, and $A_v$ is the intersection between the buffer and non-greenhouse area.

To use LEI, a buffer distance must be defined, and this was set to 30 m according to the spatial resolution of remote sensing data. A new greenhouse patch is infilling when LEI is between 50 and 100, edge-expansion when LEI is between 0 and 50, and outlying when LEI is zero. Furthermore, to reflect the aggregate properties of the grown greenhouse patches, two variant of LEI called mean expansion index (MEI) and area-weighted mean expansion index (AWMEI) were proposed by:

$$MEI = \sum_{i=1}^{N} \frac{LEI_i}{N} \tag{6}$$

where $LEI_i$ is the LEI value for a new greenhouse patch, and $N$ is the total number of newly grown greenhouse patches. Larger values of MEI indicate more compact greenhouse growth trends.

$$AWMEI = \sum_{i=1}^{N} LEI_i \times \left(\frac{a_i}{A}\right) \tag{7}$$

where $a_i$ is the area of this new greenhouse patch, and $A$ is the total area of all these newly grown greenhouse patches. Similar to MEI, a more compact growth trend of greenhouse leads to a larger AWMEI value.

## 2.8. Spatial Entropy Measure

The concept and measure of entropy have been used with different meanings in various applied sciences because of its ability to synthesize different concepts such as information, surprise, uncertainty, heterogeneity, contagion [59]. Many scientific domains, such as geography, ecology, biology, image analysis or landscape studies, which including spatial data are often required for evaluating data heterogeneity or qualifying the distribution of 'things' in space [60]. For these reasons, entropy was often applied for spatial data description and interpretation. In this study, Shannon's entropy that measures diversity in categorical data was used to measure the extent of greenhouse expansion with the integration of remote sensing and GIS, and can be defined as:

$$H(X) = \sum_{i=1}^{I} p(x_i) log\left(\frac{1}{p(x_i)}\right) \tag{8}$$

where $X$ be a discrete variable taking values $x_i$ (in this study $X$ classified each maps in $x_0$ (refers to non-greenhouse) and $x_1$ (refers to greenhouse)), while $p_X = (p(x_0), p(x_1))$ be the probability mass function (pmf) of $X$. It quantifies the average amount of information brought by $X$ based on pmf. when the distribution of classes is uniform, it will reach the maximum value ($log(I)$). In other words, lower Shannon's entropy values indicate aggregated or compact greenhouse expansion, while higher the value or closer to the maximum value indicates the dispersed or sparse expansion of greenhouses.

## 3. Results

### 3.1. Multi-Temporal Greenhouse Maps and Their Accuracy Assessment

The multi-temporal greenhouse maps were produced using the RF supervised classification for the years 1990, 1995, 2000, 2005, 2010, 2015 and 2018. The overall accuracy of all years was greater than 90%, and for years 1995, 2005, 2010 and 2018, their accuracies are even better (above 95%). The detailed accuracy assessments shown in Table 2 demonstrated that the mapping results of multi-temporal greenhouses could meet the requirements for quantifying its spatiotemporal dynamics in this study. Classification errors mainly occurred in greenhouses, white factory buildings and salt pans due to their similar regular shapes and spectral characteristics. For better visualization of the mapping results of multi-temporal greenhouses, an illustration of an example in both seasons and its classification results for a long-temporal sequence is shown in Figure 5.

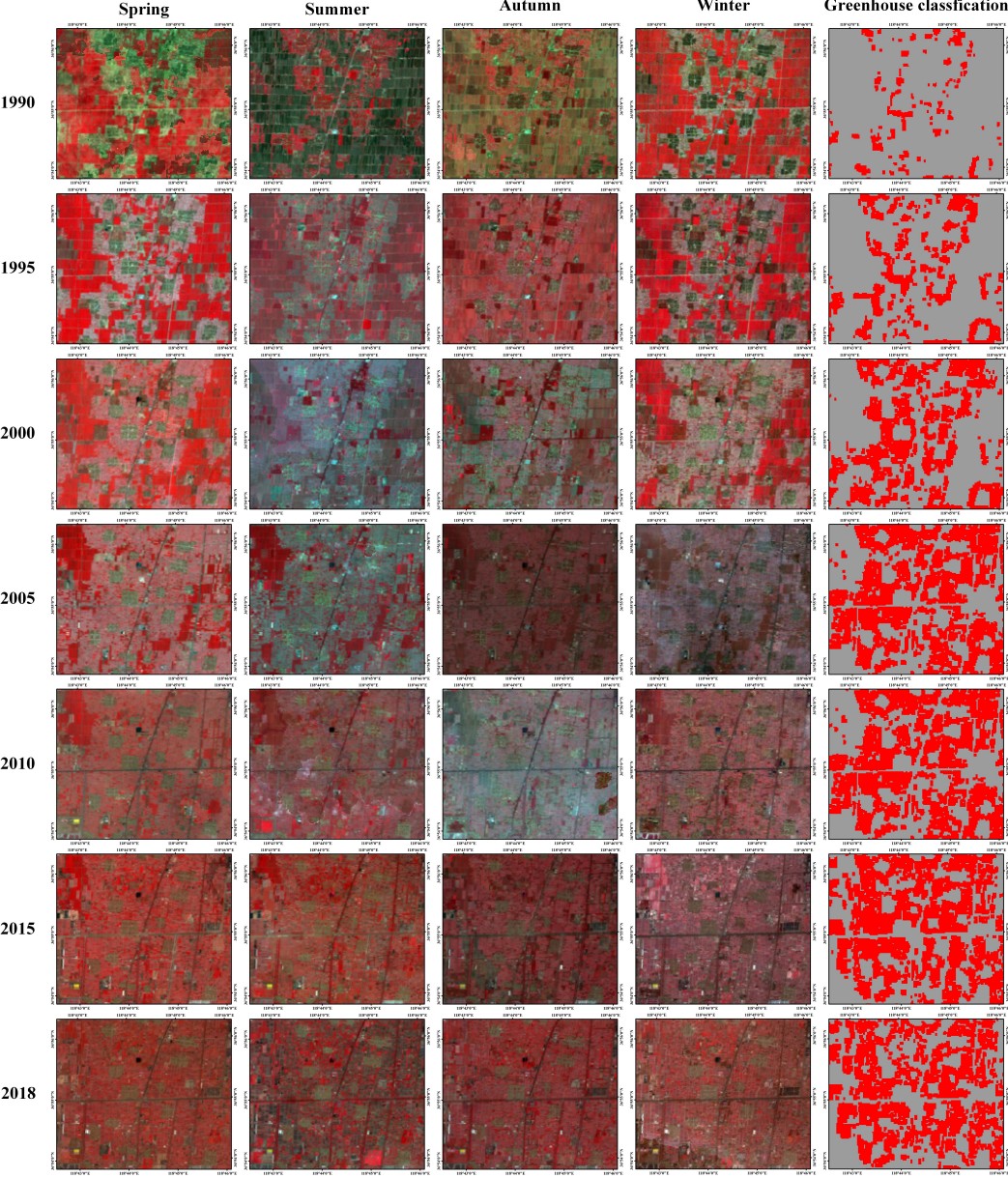

**Figure 5.** Example of landsat images that were displayed in both seasons and greenhouse classification results (in false color composite: R = NIR, G = Red, B = Green).

**Table 2.** Accuracy assessment of multi-temporal greenhouse maps.

| Years | Overall Accuracy (%) | Kappa Coefficient | Producers Accuracy (%) | Consumers Accuracy (%) |
|-------|----------------------|-------------------|------------------------|------------------------|
| 1990 | 93.28 | 0.866 | 87.42 | 88.74 |
| 1995 | 96.18 | 0.923 | 93.80 | 92.79 |
| 2000 | 93.16 | 0.864 | 88.47 | 88.44 |
| 2005 | 96.93 | 0.939 | 96.77 | 96.77 |
| 2010 | 96.91 | 0.938 | 95.59 | 95.39 |
| 2015 | 94.13 | 0.883 | 89.72 | 90.40 |
| 2018 | 96.51 | 0.930 | 95.21 | 95.58 |

We further mapped the expansion of greenhouses from 1990 to 2018 to reflect the temporal sequence of greenhouse expansion (Figure 6). For better visualization of the expansion process, pixels that converted to greenhouses earlier were given brighter colors, while those that converted to greenhouses later were given darker colors. Figure 6b–d illustrated its expansion process in an enlarged area by comparing the actual TM images in 1990 and 2018 with its temporal density.

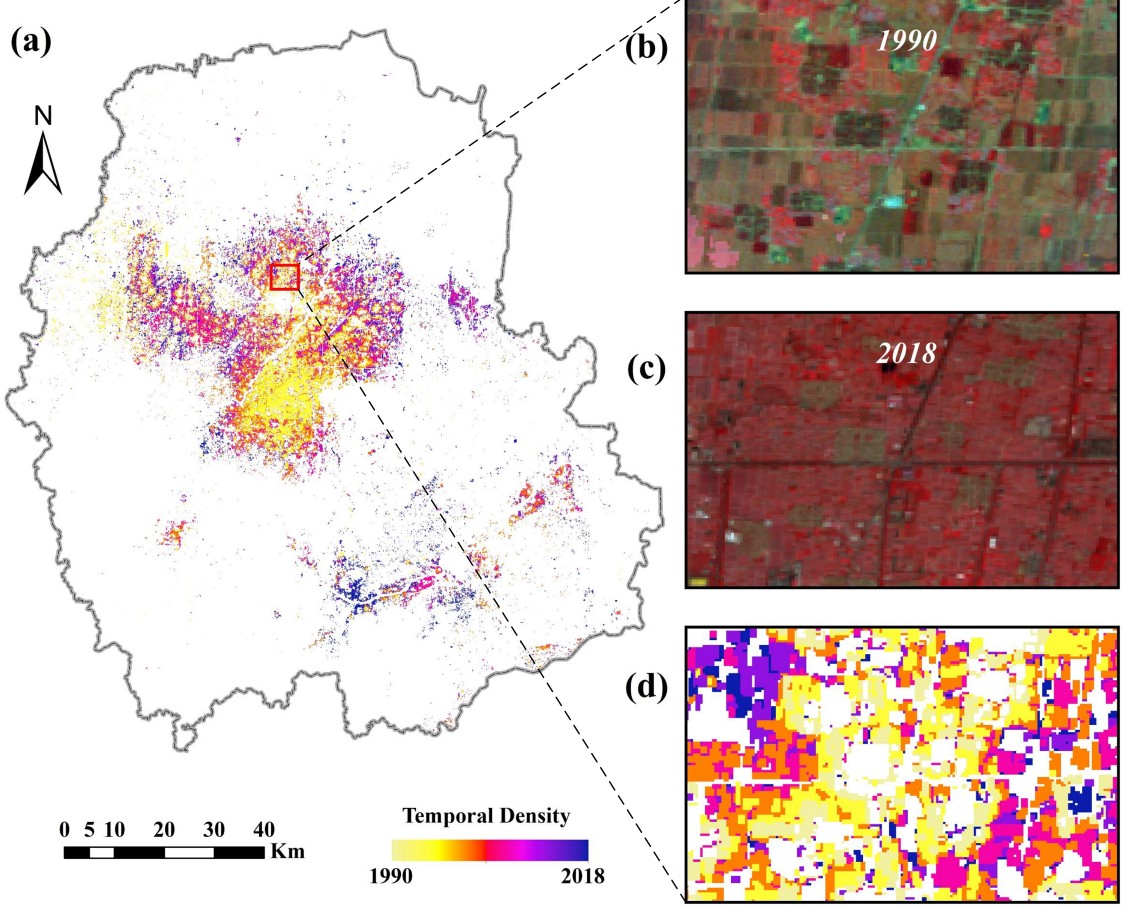

**Figure 6.** Temporal density of greenhouses expansion (1990–2018). (**a**) Overall distribution of greenhouse in temporal, (**b**) TM image acquired in 1990 (in false color composite), (**c**) TM image acquired in 2018 (in false color composite), and (**d**) Temporal density of greenhouses expansion in an enlarged area.

### 3.2. Area Change of Greenhouses during 1990–2018

According to the multi-temporal greenhouse maps, the total area of the greenhouses in the study area has increased continuously during the 28-year period, from 32.42 km$^2$ in 1990 to 1094.36 km$^2$ in 2018. The net area increase of greenhouses was about 1061.94 km$^2$ with an overall rate of 37.93 km$^2$/year during 1990–2018. The ACR for each period were 26.41 (1990–1995), 73.31 (1995–2000), 41.09 (2000–2005), 47.28 (2005–2010), 6.41 (2010–2015), 29.95 (2015–2018). Therefore, we can divide it into three periods (Figure 7), including the original period (1990–1995), the rapid expansion period (1995–2010), and the steady growth period (2010–2018).

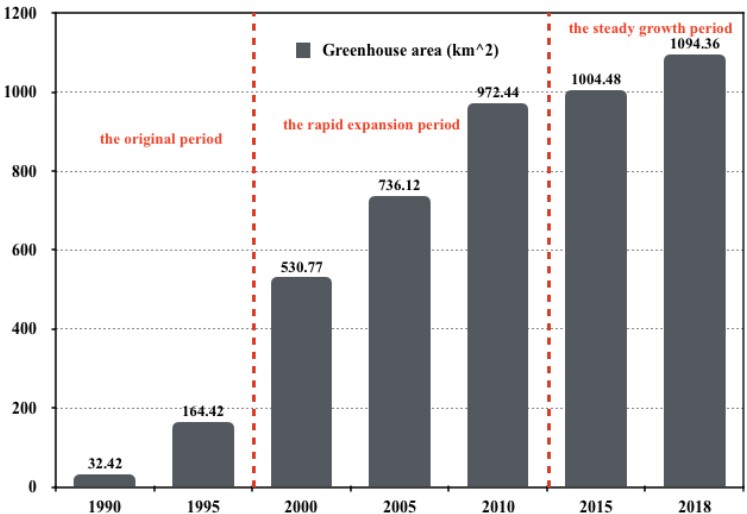

**Figure 7.** The total area of the greenhouses for each intervals.

The total area changes of greenhouses in different Direction-based buffer zones were shown in Figure 8a. The greenhouses in all directions increased over the 28 years and the largest net increase was concentrated within the 10–35 km NW buffer (268.43 km$^2$), accounting for 25% of total increased greenhouses during 1990–2018 in the study area. 77% of increased greenhouses in NW were observed in the 10–30 km buffer, 67% of increased greenhouses in SE were observed in the 35–55 km buffer and 79% of increased greenhouses in SW were observed in the 5–20 km buffer. We believe that the main reason for the significant differences in the development of greenhouses in all directions is that they are restricted by terrain and urban construction. Specifically, the northern plains are more suitable for large-scale greenhouse expansion than in the southern hilly areas. In addition, urban construction in Qingzhou, Linzi (SW direction) and Weicheng (SE direction) also affected the development of greenhouses in the study area.

Furthermore, the increased greenhouse in different Distance-based buffer zones was shown in Figure 8b. 61% of the increased greenhouse were occupied within the 10–20 km town centers buffer, 71% of the increased greenhouse were occupied within the 0–2.5 km rural settlements buffer, 42% of the increased greenhouse were occupied within the 0–5 km main rivers buffer and 94% of the increased greenhouse were occupied within the 0–5 km primary roads buffer. It illustrated that as a controlled environmental agricultural production mode, the convenience of production and the improvement of transportation conditions are the decisive factors for the site selection of new greenhouses.

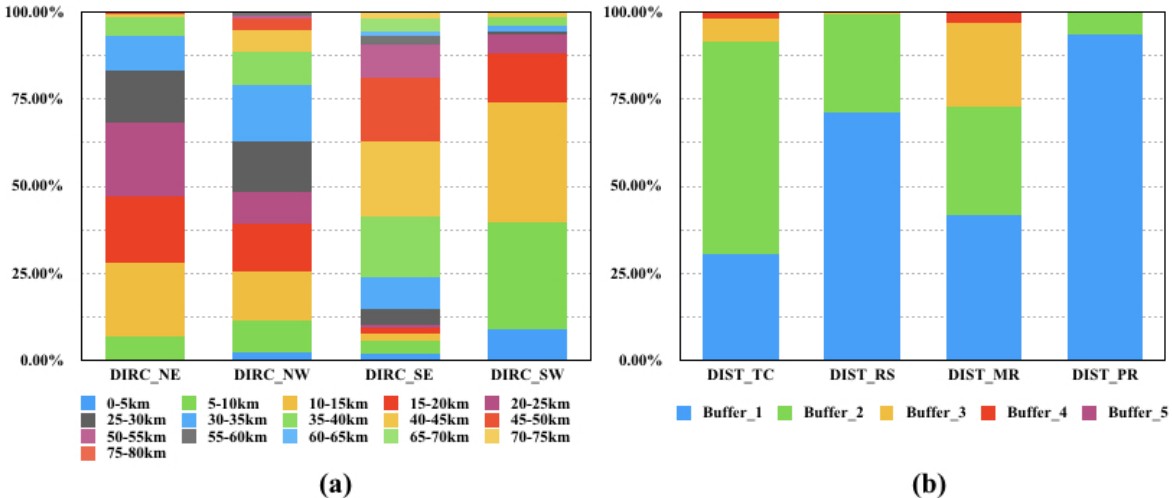

**Figure 8.** The total area changes of greenhouses in (**a**) Direction-based buffer zones, and (**b**) Distance-based buffer zones.

Given that multi-temporal greenhouse maps already being collected as polygon data in GIS, we calculated three global metrics for spatial change properties of each greenhouse patch. The NumRatio in Figure 9 showed that a large increase occurred in 1990–2010 and a rapid decline after 2010, which illustrated that the size of newly grown greenhouse patches was bigger than before with the continuous development of protected agriculture. The AreaRatio in Figure 9 showed that positive changes in the study area occurred in all years except 2015–2018 and a large increase occurred in 1995–2005, which illustrated that the growth rate of the greenhouse area in the whole study area slowed down significantly after 2015. The AvgAreaRatio in Figure 9 showed that it is below 1.00 in 1990–2010 and greater than 1.00 after 2010, which illustrated that the growth rate of the total area of the greenhouse patches was lower than the number of greenhouse patches in the early stage, but it was reversed in the later stage. And such a trend can also be obtained by observing the significant differences between NumRatio and AreaRatio (shown by the red dashed line in Figure 9) at this stage of 2005–2010.

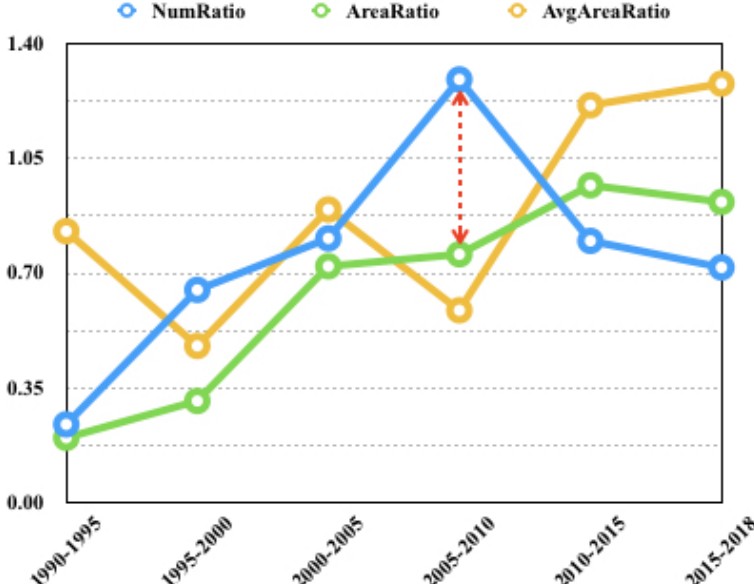

**Figure 9.** Global change metrics of greenhouses for each period.

*3.3. Landscape Metrics of Multi-Temporal Greenhouse Maps*

Results of the landscape metrics in class level for the multi-temporal greenhouse maps are presented in Table 3. In terms of the NP, there was an increase that occurred in all years except 2005–2010, which may due to the fastest expansion of greenhouse in 2005–2010 thus made the greenhouse patches more connected. The result of ED showed that a continuous increase occurred in all years, which illustrated that the extent to which their boundaries are cut apart by other landscape types is also increasing with the continuous expansion of greenhouses. The result of LSI generally showed a rising trend, which indicated that the landscape shape of the greenhouse tends to be discrete. Similar to ED and LSI, AWMPFD also showed an increasing trend, which proved that the spatial shape of the greenhouse patches is becoming more and more complicated from the fractal theory. The result of AI has remained relatively stable (greater than 80%), which indicated that the distribution of greenhouses has always had a high degree of compactness from a global perspective.

**Table 3.** Landscape pattern metrics of multi-temporal greenhouse maps.

| Year | NP | ED | LSI | AWMPFD | AI |
|------|------|--------|---------|--------|--------|
| 1990 | 959 | 0.553 | 34.161 | 1.054 | 80.379 |
| 1995 | 3987 | 2.584 | 71.115 | 1.072 | 81.635 |
| 2000 | 6051 | 5.711 | 87.554 | 1.168 | 87.390 |
| 2005 | 7434 | 7.323 | 95.285 | 1.188 | 88.331 |
| 2010 | 5672 | 7.448 | 84.399 | 1.231 | 91.022 |
| 2015 | 7112 | 8.749 | 97.527 | 1.211 | 89.771 |
| 2018 | 9953 | 10.930 | 116.716 | 1.201 | 88.258 |

Results of the landscape metrics of the increased greenhouses for Direction-based buffer zones are presented in Figure 10. These plots display the landscape metrics at the local level in order to represent the irregular buffer ranges and directions from the central pixel of the study area. In terms of the DIRC_NE, the increased greenhouses with high fragmentation (with higher value of NP and ED), complicatedness (with higher value of LSI and AWMPFD) and aggregation (with higher value of AI) appeared in 10–30 km (Buffer_3–6), 10–35 km (Buffer_3–7) and 0–25 km (Buffer_1–5), respectively, and these metrics of NW showed a similar landscape change trend with NE. Differ from NE and NW, the result of DIRC_SE showed that the increased greenhouses with high fragmentation, complicatedness and aggregation appeared in 35–50 km (Buffer_8–10), 35–45 km (Buffer_8–9) and 40–45 km (Buffer_9), respectively. In particular, there was a significant difference in landscape change trend (both in fragmentation, complicatedness and aggregation) between SE and SW. Compared with the results of SE, the spatial distribution of increased greenhouses with high fragmentation, complicatedness and aggregation in SW shrank about 25, 30 and 40 km, respectively.

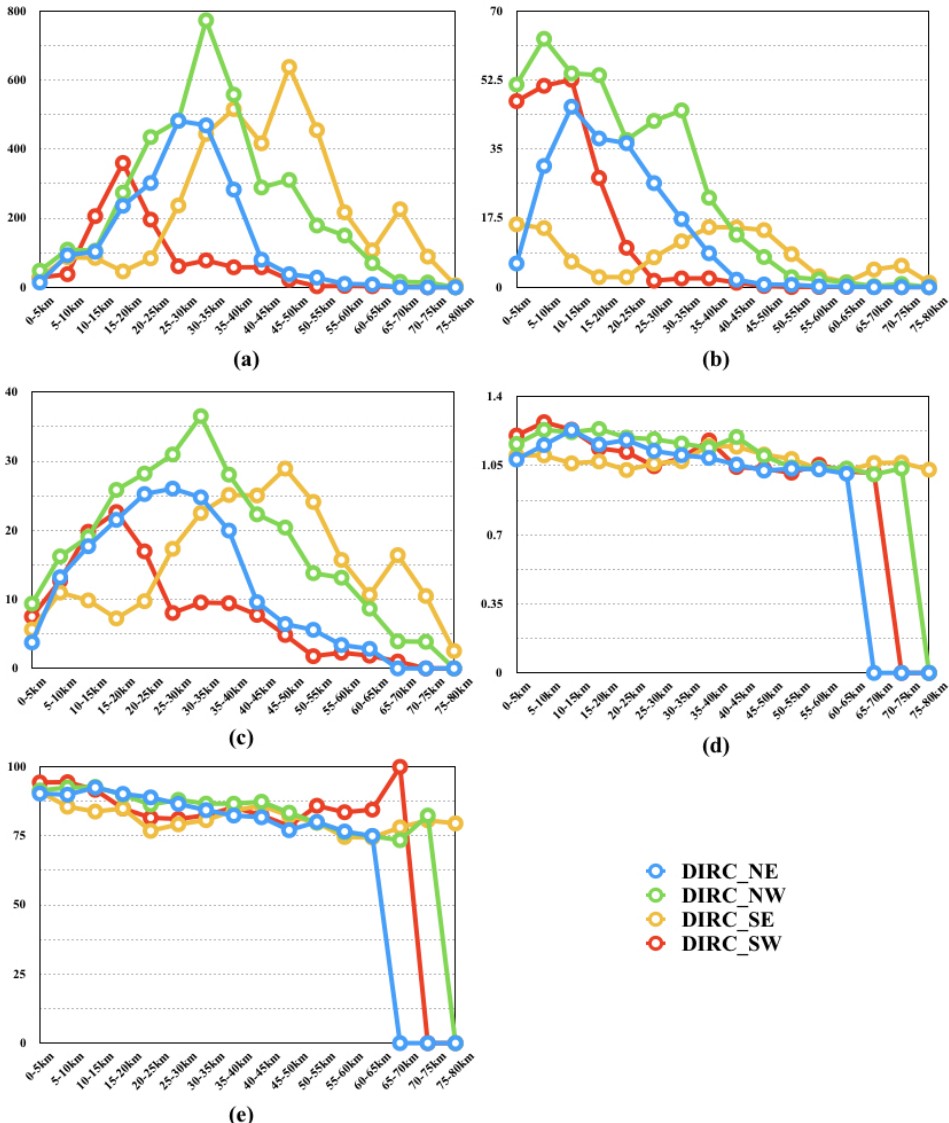

**Figure 10.** Five landscape metrics of increased greenhouses in Direction-based buffer zones, (**a**) NP, (**b**) ED, (**c**) LSI, (**d**) AWMPFD, and (**e**) AI.

Regarding the landscape metrics of the increased greenhouses in different Distance-based buffer zones, the trend of various landscape indices in different types of buffers also showed significant differences (as shown Figure 11). In terms of the DIST_TC, the trend of NP, ED, AWMPFD and LSI changes with buffer zones was similar, that is, from Buffer_1 increased to Buffer_2 and then gradually decreased. And this showed that the increased greenhouses within 10–20 km from the town centers are the most fragmented and complicated greenhouses. The result of DIST_RS generally showed that there was a great change that occurred in Buffer_3 for all the landscape metrics. And this means that the buffer zone of rural settlements in 5–7.5 km is the dividing line for the change of landscape pattern of the increased greenhouses. As for DIST_MR, ED, AWMPFD and AI appeared a similar trend of change, which slowly increaed from Buffer_1 increased to Buffer_3 and started to decline. This illustrates that the landscape pattern of the increased greenhouses in the range of 10–15 km away from the main rivers is the most complicated, and the degree of aggregation is also the highest. The result of DIST_PR showed an interesting phenomenon, that is, compared with other types of buffers, there was a significant increase of AI in from Buffer_4 to Buffer_5. This may be due to the emergence of some professional agricultural greenhouse bases far away from the town centers and primary roads.

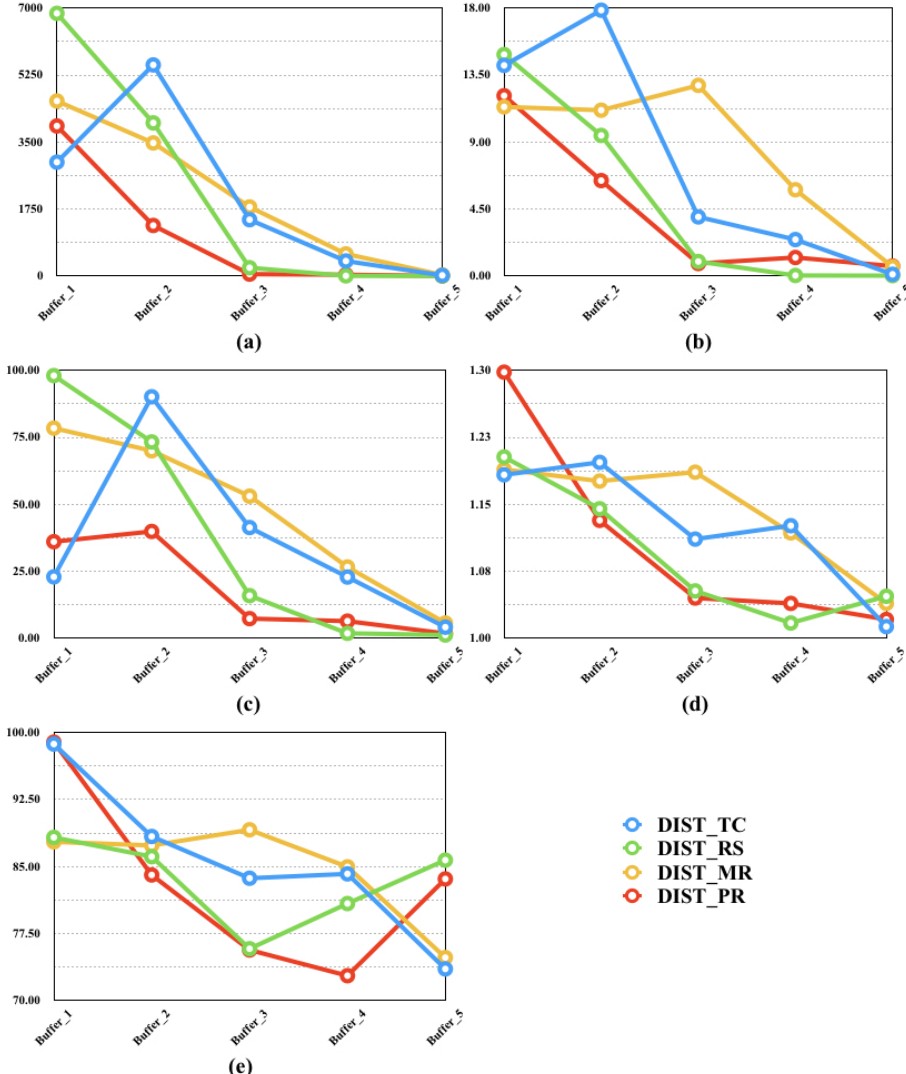

**Figure 11.** Five landscape metrics of increased greenhouses in Distance-based buffer zones, (**a**) NP, (**b**) ED, (**c**) LSI, (**d**) AWMPFD, and (**e**) AI.

### 3.4. Greenhouse Expansion Modes of Each Periods

As mentioned above, here we considered three expansion modes of the greenhouse: infilling, edge-expansion and outlying. Figure 12 presents the spatial distribution of different greenhouse expansion modes in six periods, which were identified by the LEI values. The statistical results of different greenhouse expansion modes including percentages of area and number of patches are also presented in Figure 13. As can be seen from the Figure 12a, the expansion mode of greenhouses was mainly outlying in the first period (1990–1995). With the new greenhouse patches that emerged, the pattern of greenhouses expansion had changed since 1995. During the rapid expansion period (1995–2010), the dominant expansion mode of greenhouses was edge-expansion (Figure 12b–d). And it is particularly noteworthy that, there was a clear outbreak of the infilling mode in the late stage of this period (Figure 12d). In the last two periods (2010–2018), as the expansion rate of greenhouses has decreased, the main expansion modes of the greenhouse were edge-expansion and infilling that took place along with old greenhouse patches (Figure 12e,f). In terms of the quantization results of MEI and AWMEI (Table 4), the smaller MEI and AWMEI values in the first period (1990–1995) also illustrated the relatively higher dominance by outlying expansion, while the larger values of MEI and AWMEI values after 1995 indicated a more compact expansion trend.

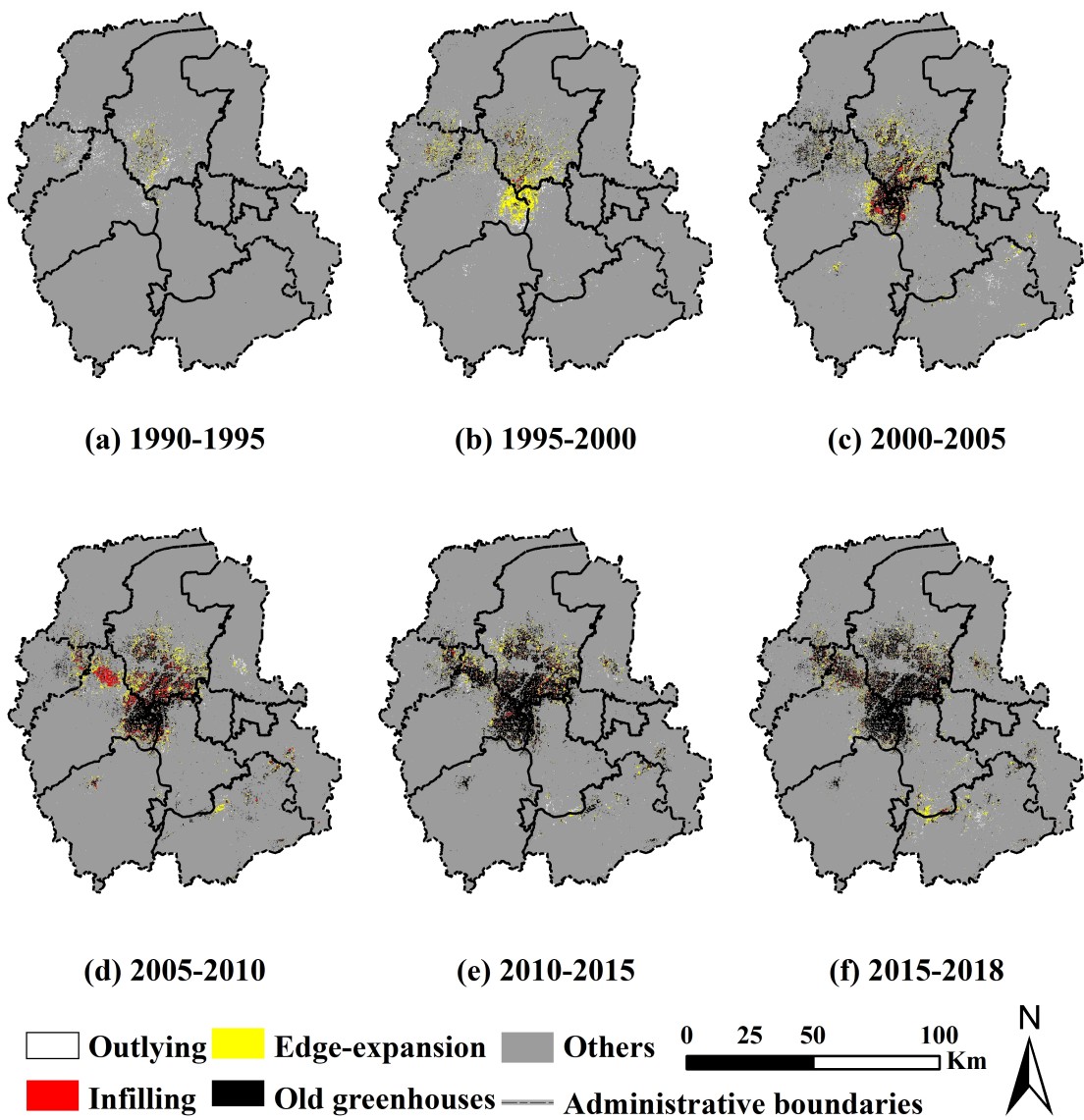

**Figure 12.** Spatial distribution of different greenhouse expansion modes in six periods.

**Table 4.** MEI and AWMEI of newly grown greenhouses patches in the six periods in study area.

| Period | 1990–1995 | 1995–2000 | 2000–2005 | 2005–2010 | 2010–2015 | 2015–2018 |
|--------|-----------|-----------|-----------|-----------|-----------|-----------|
| MEI | 7.130 | 19.859 | 31.398 | 41.356 | 37.982 | 35.783 |
| AWMEI | 5.180 | 19.029 | 25.244 | 36.004 | 26.487 | 22.920 |

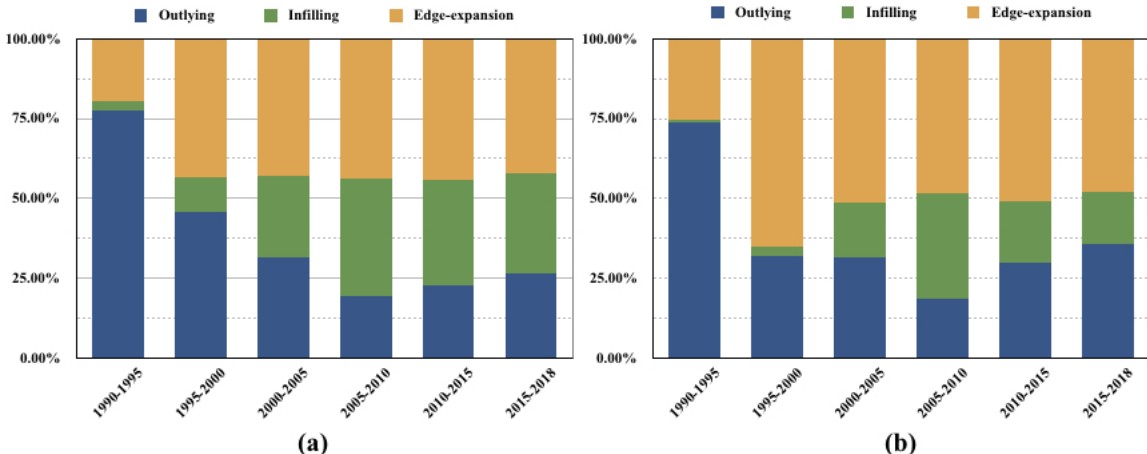

**Figure 13.** Percentages of growth area (**a**) and number of patches (**b**) for the three greenhouse expansion modes in the six periods.

## 3.5. Results of Shannon's Entropy

The relative Shannon's entropy for the years 1990, 1995, 2000, 2005, 2010, 2015 and 2018 are shown in Figure 14. The calculated Shannon's entropy was lower than 0.1 before 2000, then increased dramatically to around 0.2 during 1995–2010 and then it remained a steady growth after 2010, which showed a highly consistent with the growing trend of the greenhouse area.

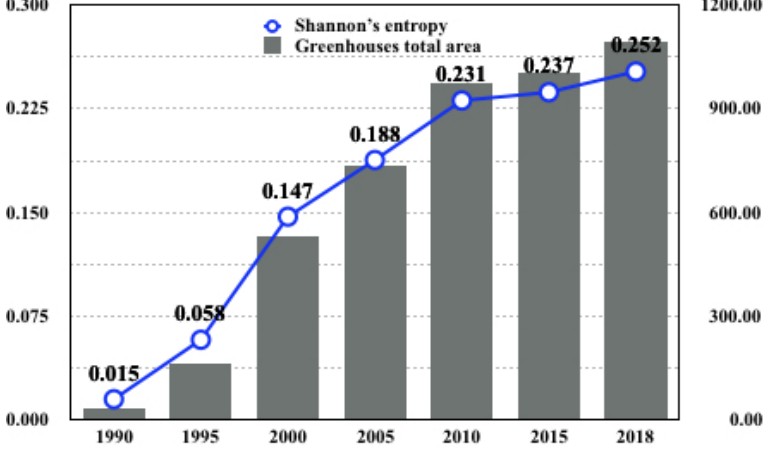

**Figure 14.** Shannon's entropy in seven intervals.

Similar to the area changes analysis and landscape pattern change analysis, we also overlapped the increased greenhouses from 1990 to 2018 in different Direction-based and Distance-based buffer zones and calculated Shannon's entropy for each buffer zones (Figure 15). Higher Shannon's entropy values of 0.690 (NE) appeared in 10–15 km (Buffer_3) and 0.687 (NW), 0.381 (SE) and 0.664 (SW) appeared in 0–5 km (Buffer_1) showed dispersed expansion of greenhouses. However, the aggregated greenhouse expansion is noticed at the outskirts in 60–65 km (NE), 70–75 km (NW), 75–80 km(SE) and 65–70 km (SW). These results confirmed that a more aggregated greenhouse moved from the center of the study area towards the periphery in all directions. As for Shannon's entropy in different Distance-based buffer zones, the increased greenhouses with higher Shannon's entropy values were occupied within the 10–20 km town centers buffer, the 0–2.5 km rural settlements buffer, the 10–15 km main rivers buffer and the 0–5 km primary roads buffer, which reveals that dispersed greenhouse growth have appeared in nearest buffers of rural settlements and primary roads.

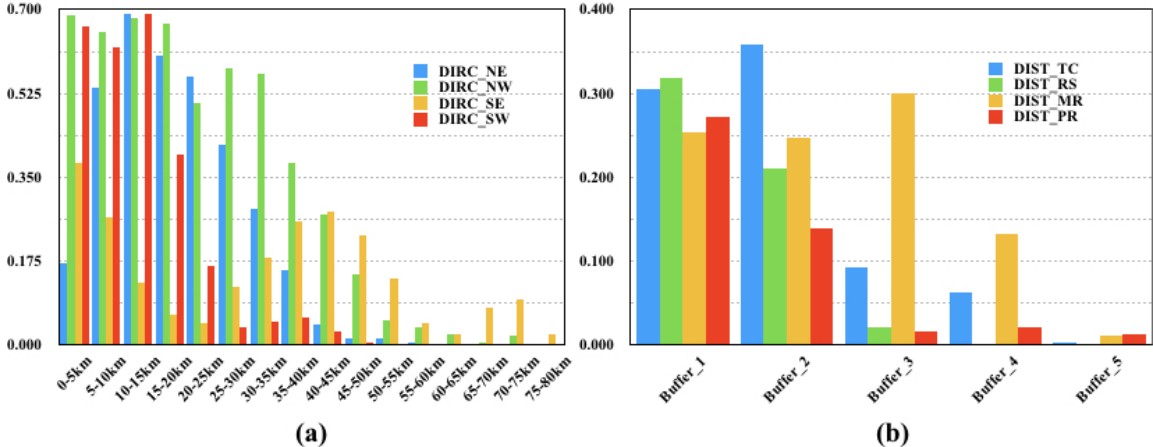

**Figure 15.** Shannon's entropy of the increased greenhouses from 1990 to 2018 in different buffers, (**a**) Direction-based buffers, (**b**) Distance-based buffers.

## 4. Discussion

### 4.1. Advantages and Limitations of Multi-Temporal Greenhouses Mapping in GEE

The proposed multi-temporal greenhouses mapping method used the RF supervised classification algorithm with a series of indices and topographic data within the GEE platform. With the high-speed image processing tools and online non-commercial remote sensing data service of GEE, we have successfully completed multi-temporal greenhouses classification based on multi-season Landsat images, which is difficult to achieve for desktop processing. And as shown in Section 3.1, we got the fine-mapping results for all the years that could meet the requirements for the spatiotemporal analysis in this study. Although it showed great potential in dealing with massive image processing for multi-temporal classification in GEE, there are also some limitations to the use of this approach. Firstly, it is unable to perform image segmentation that covering such a large area in GEE at present [61], which limits our use of object-oriented methods for greenhouse classification. Secondly, unbalanced reference data of each year due to the lack of the historical high-resolution aerial may lead to instability of classification accuracy. Furthermore, some systematic errors such as user memory limit exceeded or internal error occur in GEE, which has a certain impact on work efficiency.

### 4.2. Analysis of the Spatiotemporal Dynamics of Greenhouses

Based on the conducted multi-temporal greenhouses maps, quantitative analysis of the spatiotemporal dynamics of greenhouses were presented in several aspects, including area changes analysis (Section 3.2), landscape pattern change analysis (Section 3.3), spatial modes of landscape expansion (Section 3.4) and spatial entropy measures (Section 3.5). The results from the analysis of the spatiotemporal dynamics of greenhouses highlight four key points.

First, the results of ACR confirmed that the rapid expansion of greenhouses in the study area occurred in 1995–2010, especially from 1995–2000. Meanwhile, the results of the increased greenhouses in Direction-based buffer zones showed that the expansion of greenhouses varied in all directions, especially when comparing the northern and southern areas, which mainly affected by terrain and policy factors. And the nearest buffer zones of rural settlements, primary roads contained most of the increased greenhouses, which also confirmed the importance of production convenience and transportation. Additionally, global change metrics of greenhouses for each period revealed that there are significant differences between the total area growth and the number of greenhouse patches growth in the process of greenhouse expansion. Combined with the local historical materials, we learned that when the 'winter-warm' greenhouse was first introduced in 1989, the development of greenhouse in the study area was only the spontaneous behavior of local farmers in pursuit of interests.

By 1995, with the growing scale of the development of the greenhouse, the local government began to implement a series of policies to promote the development of this industry, including adjusting the agricultural planting structure, opening 'green channels' and financial support. It can be seen that our study elaborated on the whole process of greenhouses development from another perspective.

Second, our study explored the characteristics of landscape pattern in each year, in which we found that the spatial configuration and compactness of the greenhouses has significantly changed as a result of the continuous growth of greenhouses. At the same time, we found a similar landscape change trend between NE and NW from Direction-based buffer zones and various results from Distance-based buffer zones. According to the literature review, we found that the development model of the greenhouses in the study area has evolved from independent construction to standardization, basement and branding. Therefore, our study quantified such a process from the perspective of landscape ecology.

Third, a new landscape index for quantifying greenhouse expansion was introduced in this study, which unveiled different expansion modes of greenhouses and identified the expansion processes of diffusion and coalescence in every period. From 1990 to 1995, outlying growth increased more prominently and became the main type of greenhouse expansion. After 1995, greenhouse development along old greenhouse patches was prominent, and edge-expansion growth became dominant. From 2010 to 2018, vacant non-developed greenhouse areas were filled inwards, making greenhouse land more aggregated and compact. We infer that these changes were caused by modes of greenhouse vegetable promotions. According to local statistics, the current typical modes of greenhouse vegetable promotions in the study area mainly including 'specialized cooperative economy organization + farmer', 'leading agriculture-involved enterprise + farmer', 'family farm management', 'demonstration garden of agricultural science and technology + farmer' and 'local agricultural fair'. These different promotion modes may lead to different plans for new greenhouses.

Finally, according to the spatial entropy measures in the entire study area and various buffer zones, we can see that the spatial heterogeneity of the increased greenhouses was different between the global and local levels. The greenhouse area with higher Shannon's entropy values will be unable to efficiently allocate the necessary resources for agricultural production. On the other hand, relevant social science research shows that the level of cooperation among the business entities is relatively low, and the contradiction between decentralized operation of greenhouse farmers and large-scale basements has not been effectively solved, which has not been able to meet the long-term development needs of greenhouse, but also seriously restricts the optimization, upgrading and promotion of the protected agriculture in the future.

*4.3. Future Works*

We view the greenhouse as a typical object of Human–Environment interaction in agriculture [62], and our research aims to achieve a better understanding of the past, present, and future of it. Future work should consider several aspects, including (1) expanding the study area of the greenhouse to achieve a more macro view (e.g., at provincial or national scale) with more advanced classification approach like deep learning [63]; (2) seeking deep insight into market demand, government planning, human decision and their consequences on the greenhouse by connecting socio-economic data with our spatial data using Meta-analysis [64]; (3) simulating the greenhouse dynamics by coupling top-down (e.g., agent-based model [65]) and bottom-up (e.g., cellular automata [66]) strategies for different hypothetical scenarios in the future.

**5. Conclusions**

In this study, we first conducted multi-temporal greenhouse maps and explored the spatiotemporal dynamics of greenhouses in the Shouguang region, north China from 1990 to 2018. Compared with traditional social science research, our study substantially advances the understanding

of the spatial distribution, process and status of greenhouses in such a typical protected agriculture region from a novel way.

By using the RF algorithm with Landsat imagery derived from the Google Earth Engine, we mapped the distribution of greenhouse with overall accuracies of 93.3%, 96.2%, 93.2%, 96.9%, 96.9%, 94.1%, and 96.5% for each year. According to the total area changes results of multi-temporal greenhouse maps, greenhouses in the study area significantly expanded during 1990–2018 by 1061.94 km$^2$ with an overall rate of 37.93 km$^2$/year. 25% of total increased greenhouses occurred in 10–3 km northwest buffer zone and 94% of the increased greenhouses were occupied within the 0–5 km primary roads buffer. Results of global change metrics showed that there was a significant difference between the growth rate of greenhouse total area and the number of greenhouse patches, especially in the period of 2005–2010. From the perspective of landscape change pattern, the fragmentation, complexity and compactness vary from different periods and buffer zones. In addition, we found that the early development of greenhouses was dominated by outlying mode, and the latter was dominated by infilling and edge-expansion modes. Although the global Shannon's entropy showed a highly consistent with the total area of greenhouse in each period, the spatial heterogeneity at the local level was not always related to the total area in each buffer zones. Our conclusions provide useful information and scientific guidance for efficiently planning and managing protected agriculture in such a typical region.

**Author Contributions:** C.O. and J.Y. conceived the study design, developed the models and drafted the manuscript. J.Y. also provided the funding. C.O. and Z.D. was involved in data acquisition and analysis and worked on aspects of the experiment evaluation. Y.L. and Q.F. improved the conceptual framework and updated the manuscript. D.Z. was involved in the design and revision of the manuscript. All authors have read and agreed to the published version of the manuscript.

**Funding:** This research was jointly funded by Special Fund for Scientific Research on Public Causes (201511010-06) and China Postdoctoral Science Foundation (2018M641529, 2019T120155).

**Acknowledgments:** The authors would like to thank the google earth engine (GEE) team and the user community for their useful feedback during this research process. And thank the journal's editors and anonymous reviewers for their kind comments and valuable suggestions to improve the quality of this paper.

**Conflicts of Interest:** The authors declare no conflict of interest.

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
