# Peer review of "Long-Term Mapping of a Greenhouse in a Typical Protected Agricultural Region Using Landsat Imagery and the Google Earth Engine"

_remotesensing, doi:10.3390/rs12010055_

Round 1

Reviewer 1 Report

In this study, a long term mapping for greenhouses were conducted in the Shouguang region, north China from 1990 to 2018. Authors used the archived Landsat satellite images on google earth engine with the random forests algorithm. The overall accuracies were over 90% for all studied periods on mapping the distribution of greenhouses compared with the ground truth data. In general, the manuscript is interesting and well written with adequate description for each manuscript section.  The result is promising for mapping greenhouses and its temporal dynamics. However, I have the following comments:

Title: The title is long and the term (Long-term mapping) give the same meaning for (Spatiotemporal Dynamics Quantifying) in my opinion. I suggest removing the part (and its Spatiotemporal Dynamics Quantifying).

L4: Remove (However).

L35-36: Use km instead of hm.

L86-89: Not understandable, rephrase.

L 118: Explain the SLC-off in a short sentence.

L123: There is no 30 February.

L124: Define the term (Pixel-qa).

L125: You already cited this reference (28).

L141: How did you collect these polygons? What is the shape and area of these greenhouses? Are the shapefiles include greenhouses only or surrounding area?

L142: What are the other 504 polygons?

L143: Why 10000 points?

L147: Are the high-resolution images that you used were from the same years (1990, 1995, 2000, 2005, 2010, and 2015) or years between? Is it one image per year or more?

L149: Are the (records of local historical agricultural statistics) a georeferenced data? What kind of historical data it was (points, polygons or paper maps or just tables with address)?

L170: (global level), Do you mean county level?

Figure 9: How you calculated these parameters? In another way, what is t1 and t2 for each value? Is it the current time with the previous stage or you have a datum year? You can clarify this at lines 186-190 in your materials and methods section.

L359: Give an example for the GEE systematic errors.

L405: (relevant), small letter.

L589: Where is the sample. Revise it.

Suggestion: I suggest to add the GEE code as it becomes a common trend for publications on GEE applications. It is just a suggestion and the authors can decide if they want to share it or no.

Author Response

Thank you for the comments which are highly insightful and enabled us to improve the quality of our manuscript. In the following pages are our point-by-point responses to each of the comments.

Reviewer 2 Report

The manuscript titled "Long-Term Mapping of Greenhouse and Its Spatiotemporal Dynamics Quantifying in a Typical Protected Agricultural Region Using Landsat Imagery and Google Earth Engine" presents a study that develop and test advanced image classification techniques with Landsat time-series in Google Earth Engine for multi-temporal greenhouse mapping in a typical protected agriculture region in China and quantify their spatiotemporal characteristics.

The paper is very interest, well written and easy to read though needs very few revisions.

Very few points are listed below:

L123: cancel “/30” near February,

L288, L291, L296, L326: after the comma, use lowercase in the text;

Figure 14: insert the unit of measurement near “Total area”, I suggest to write “greenhouses total area”;

L397 and L406: I suggest to include some references about the “local statistics” and “social science research”.

Author Response

(The authors gave the same response as above.)

Reviewer 3 Report

This paper deals with the use of Landsat time-series to analyse multi-temporal greenhouse evolution in a specific region of China. A priori, it may seem that this issue has already been addressed in many other researches. The particularity of this study is that the classification is performed using Random Forests (RF) based on the Google Earth Engine (GEE) platform. The results of classification are assessed by quantifying a range of spatiotemporal dynamics: global change analysis, landscape pattern metrics, expansion indicators and spatial entropy parameters. The results are statistically sound, and the authors perform a detailed discussion on greenhouses evolution (from 1990 to 2018 with a 5-year intervals) based on them. The manuscript is written with an organized structure of research paper and in plain English. Only minor spell check is required in some parts of the text (e.g. lines 296, 326 [*the], 380, etc.).

In my opinion, the topic clearly matches with the scope of Remote Sensing and it could be an interesting contribution for the readers. Thus, I think that it could be approved for publication, but I suggest some previous corrections (in no particular order):

Line 47-50. I am not sure that reference [13] can be considered a mono-temporal study. Line 161. The authors use RF with 500 trees, which is a high number, even for a huge dataset. Indeed, the more trees the more reliable results. However, at a certain point, the benefit in prediction performance from learning more trees will be lower than the cost in computation time for learning these additional trees. In this regard, I would have liked to see in the article some additional comment on the performance of GEE and the processing times. In my opinion, this is the cornerstone of the paper and may justify the interest of the research using the computing power of GEE. Therefore, it should be discussed in more detail. Line 133. It is unclear how authors use SRTM data (elevation and slope) to “distinguish some misclassified objects”. I would suggest to edit the axis labels of plots that include direction-based buffer zones (Figures 8a,10, and 15a) indicating their specific values (in km) instead of buffer_1,… buffer_16, for a direct reading and to aid in disambiguation with distance-based buffers buffer_1,… buffer_5. Line 359. “some systematic errors often occur in GEE, which has a certain impact on work efficiency”. Please clarify.

Author Response

(The authors gave the same response as above.)
